# Cortisol in Manure from Cattle Enclosed with Nofence Virtual Fencing

**DOI:** 10.3390/ani12213017

**Published:** 2022-11-03

**Authors:** Christian Sonne, Aage Kristian Olsen Alstrup, Cino Pertoldi, John Frikke, Anne Cathrine Linder, Bjarne Styrishave

**Affiliations:** 1Department of Ecoscience, Aarhus University, Frederiksborgvej 399, DK-4000 Roskilde, Denmark; 2Department of Clinical Medicine, Aarhus University, Palle Juul-Jensens Boulevard 165, DK-8200 Aarhus, Denmark; 3Department of Nuclear Medicine and PET, Aarhus University Hospital, Palle Juul-Jensens Boulevard 99, DK-8200 Aarhus, Denmark; 4Department of Chemistry and Bioscience, Section of Bioscience and Engineering, Aalborg University, Fredrik Bajers Vej 7H, DK-9220 Aalborg, Denmark; 5Aalborg Zoo, Mølleparkvej 63, DK-9000 Aalborg, Denmark; 6Wadden Sea National Park, Havnebyvej 30, DK-6792 Rømø, Denmark; 7Toxicology and Drug Metabolism Group, Department of Pharmacy, Faculty of Health and Medical Sciences, University of Copenhagen, Universitetsparken 2, DK-2100 Copenhagen, Denmark

**Keywords:** cattle, virtual fencing, manure, cortisol, stress

## Abstract

**Simple Summary:**

To increase the efficiency and geographic expansion of nature conservation, large grazers have recently been used, either in the form of wild hoof-bearing animals or as domesticated ruminants including cattle. As part of this, controlling the movement of these animals is essential using either physical or virtual fences to manage the areas of interest. Physical fencing limits migrating wildlife, while using virtual fences with GPS technology paired with collars emitting auditory and electric cues encourages the animals to stay in the desired area without physical restrictions for wild animals. However, virtual fences raise ethical questions regarding the electric impulses emitted by the collar and stress in the fenced animals, we show that the stress hormone cortisol in cow mature is not associated with the use of virtual fencing. We, therefore, conclude that there is no evidence suggesting that cows are stressed from the use of virtual fencing, thus making virtual fencing a reasonable alternative to traditional electric physical fencing for cows. We recommend using manure as a noninvasive physiological measure of large grazer stress during virtual fencing to survey and understand animal welfare.

**Abstract:**

To increase the efficiency and geographic expansion of nature conservation, large grazers have recently been used, either in the form of wild hoof-bearing animals or as domesticated ruminants including cattle. Using physical fencing limits migrating wildlife, while virtual fences encourage the animals to stay in the desired area without physical restrictions on wild animals. However, virtual fences raise ethical questions regarding the electric impulses emitted by the collar and stress in the fenced animals. Here, we tested if keeping twelve Angus cows (*Bos Taurus*) in a virtual fencing (Nofence©) compromised their welfare. For this purpose, we collected manure samples from five cows every second day prior to and after the transition from traditional to virtual fencing over a period of 18 days. Cortisol concentrations were 20.6 ± 5.23 ng/g *w*/*w* (mean ± SD), ranging from 12 to 42 ng/g *w*/*w* across individuals and concentrations did not change over the study period. We, therefore, conclude that there is no evidence suggesting that the cows were stressed from the use for virtual fencing, thus making virtual fencing a reasonable alternative to traditional electric physical fencing of cows.

## 1. Introduction

The beneficial effects of rewilding have attracted increased attention over the past decade. Among others, the European Union has, as part of their new Horizon Europe 2030 program, expressed interest in supporting this management and research field to increase habitat conservation and ecosystem services [1]. Often, rewilding using larger grazers, including ruminants and hoof-bearing mammals, helps shape the biodiversity composition of plant and vertebrate [1]. However, rewilding has been subject to public concern. The use of physical fencing is a highly discussed topic due to animal welfare and the fact that it not only encloses the animals but also limits public access to these areas [2,3,4,5,6]. Moreover, physical fences, of which some are electric, can act as a barrier to migrating wildlife and any other kind of movement including the public using nature for recreation [2,3]. In addition, physical fences are not easily adaptable, making them difficult to move in relation to various management practices that often require temporary exclusions of grazing animals from environmentally sensitive areas, e.g., during the breeding season of vulnerable ground-breeding bird species [4,5]. Nonetheless, public concern about animal welfare is a challenge when implementing virtual fencing systems in conservation grazing and rewilding projects [6]. The discussion of these concerns for recreation and biodiversity calls for more research to support virtual fencing, enabling the use of nature, and protecting and managing ecosystems and biodiversity.

Virtual fencing systems using GPS unit technology are increasingly used as an alternative to physical fencing as an efficient way of keeping animals within or outside a given area when using herbivores for flexible grazing of large areas [7,8,9,10,11]. Virtual fencing systems utilise GPS collars. Each collar is composed of a GPS, loudspeaker, and battery that collect and send information about the animal’s position, number of auditory cues, and electrical impulses emitted upon the animal approaching a virtual invisible boundary using a smartphone or tablet [12]. So far, the use of virtual fencing is not yet legal in Denmark or other European countries However, preliminary behavioural studies do not suggest poor animal welfare in this case [12]. Therefore, regarding animal welfare, there is a need for more studies on stress in relation to virtual fencing and if it affects fenced animals’, for example, stress hormones such as cortisol [12].

Stress and animal welfare are dependent on how an animal perceives its environment. Previous experiences affect the animal’s expectations of stimuli outcomes, thus contributing to the individual appraisal of a stimulus and whether a stimulus is perceived as positive or negative [13]. Individual stress relates to an animal’s learning ability to expect a stimulus response. Moreover, the predictability and controllability of a situation influence how animals learn, thereby partially determining the welfare outcomes from the situation [13,14].

Cattle and sheep welfare in virtual enclosures is comparable to that of traditional physical electric fencing mainly due to adaptation [12,14,15,16,17]. However, when assessing animal welfare, it is important to consider not only the behavioural responses and learning abilities of the animals, but also the physiological indicators of stress. The major corticosteroid hormone in mammals participating in physiological regulation and homeostasis of metabolism, growth, and development is cortisol. In addition, it influences the stress responses and the regulation of the physiology and endocrinology of the reproductive and immune systems [18,19].

Thus, one way to noninvasively measure welfare and stress in mammals is to analyse cortisol concentrations in manure. This information is lacking in the international scientific literature in terms of virtual fencing, and so the present study adds new information to the existing knowledge on animal welfare and the use of systems such as Nofence©. The aim of this study was therefore to assess the effect of a virtual fencing system (Nofence©) on the welfare of twelve Angus cows (*Bos taurus*). Here, we hypothesised that using virtual fencing does not compromise Angus cows’ wellbeing as measured by manure cortisol concentrations.

## 2. Materials and Methods

### 2.1. Location and Study Design

Twelve Angus cows aged 4–9 years were held within virtual fencing on the east coast of the island Fanø in southwest Denmark. To ensure logistics and sample quality, and reduce labor, five randomly chosen individuals of the twelve cows were selected for manure collection and subsequent analyses of cortisol. All cows were pregnant during the experiment. The total project area was 65 ha and consisted of coastal meadows to the east and a dune landscape with both dry (heath) and wet parts, with scattered vegetation of trees and bushes to the west. The maximum area used during this particular experiment was limited to 35 ha, which was encompassed in the larger used 65 ha area. This area was camera-monitored during the first weeks of virtual fencing to assess behaviour and collection manure. The area was chosen from a logistic point with access to electricity, roads, video recording, and research facilities (housing, labs a.o.). The cows were accustomed to traditional physical electric fencing, but not familiar with virtual fencing prior to the study. Changing the fence line of the virtual border was conducted step-wise to ensure a smooth learning curve. On 28 May 2021, the twelve cows were therefore placed in a physically electrically fenced training enclosure of about 6.5 hectares and, after a day and a half (29 May), a virtual border replaced the southwestern fence line to ensure sufficient grass for the cows. After six days, the border was moved 20 m further south (4 June) (Figure 1). The virtual fence was moved another 20 m each on 7 and 10 June. The remaining three sides of the physical electric fence were removed on day 14 of the experiment (12 June) (Figure 2). This particular experiment was concluded three days later (15 June), while the behavioural study conducted by Aaser et al. [12] continued another 121 days (until 14 October 2021). We selected 14 days to ensure a short window of stable weather with respect to wind, participation, and food access. Animal Experiments Inspectorate of Denmark approved the study (2020-15-0201-00588), conducted according to the institutional guidelines for animal research (directive 2010/63/EU).

### 2.2. Virtual Fencing System Nofence©

Briefly, the Nofence© collars, placed loosely around the animals’ neck, contained two small solar panels and a GPS receiver measuring the position, heading, and speed of the animals, with each collar having a unique serial number. The collar’s GPS position was continuously assessed in relation to the specified virtual boundary to determine if the animal was approaching the virtual boundary. When an animal approached the virtual boundary, the collar emitted a series of warning sounds, consisting of multiple 82 dB tones increasing in pitch for a duration of 5–20 s, based on whether the animal continued to ignore the warning or changed direction. If the animal ignored the warning and continued to approach the virtual boundary and was at risk of crossing the boundary, it received an electric impulse of 0.2 J at 3 kV for one second. The risk of an animal crossing the virtual boundary was determined by its position, heading, and speed. The animal received another electric impulse if it did not respond to the previous impulse by slowing down, changing heading, or stopping. After a maximum of three impulses, the owner received an escape message. The collar continued to monitor the location of the escaped animal, but the animal did not receive any warnings or impulses during this time. Once the animal re-entered the virtual enclosure, the collar returned to normal function. The collars continuously collected and transmitted information about warnings and electric impulses and recorded the cows’ activity every 30 min based on an accelerometer. Aaser et al. [12] provided more details.

### 2.3. Sampling

To study the potential stress response to virtual fencing, manure samples were collected from five of the twelve individuals every second day in the period of 29 May to 15 June 2021, during which the cattle learned about the virtual fencing. The reasons for sampling only 5 cows was to ensure that each faecal sample could be assigned with accuracy to an individual. Only fresh samples were collected at mid-day, because the collector had to register the cow’s ear tag, and therefore had to observe that manure was deposited for each cow. The ten samples per cow were analysed for cortisol levels to measure upregulation of the brain–adrenal–cortisol stress-signalling axis following activation of the virtual fence.

### 2.4. Cortisol Analyses

All samples were taken with a clean polyethylene (PE) sampling bag and stored frozen at −20 °C until laboratory analyses from December 2021 to March 2022. Initially, samples were freeze -dried in a freeze, and aliquots were transferred to Eppendorf tubes. We added 50 µL of internal standard containing 0.1 μg/mL of deuterated steroid analogues to each 200 mg sample followed by 1 mL extraction solvent (H_2_O/methanol, 20:80, *v*/*v*). Samples were then homogenised at 3200 rpm for one minute using a BeadBlaster^TM^ 24 Microtube homogeniser (SoCal Biomed, Lake Forest, CA, USA), followed by centrifugation at 12,000 rpm for six minutes, and supernatants were transferred to 10 mL glass vials. An additional 1000 µL of extraction solvent (H_2_O/methanol, 20:80, *v*/*v*) was added to the Eppendorf tubes, and the samples were again homogenised in a BeadBlasterTM followed by centrifugation. This procedure was repeated three times in total. Then, 450 µL of acetate buffer (pH = 5.0 ± 0.1) and 50 µL of Β-glucoronidase/arylsulphase (Roche Diagnostics, Mannheim, Germany) from Helix pomatia were added to each sample [20,21]. Deconjugation was conducted by incubating samples at 37 °C for two hours. We prepared 0.4 M acetate buffer from sodium acetate solution and acetic acid, which we adjusted to pH = 5.0 ± 0.1.

Following extraction, the glass vials containing the collected supernatants were then diluted to 9.0 mL by adding 6.0 mL of MilliQ-water. The samples were stored at −18 °C until solid-phase extraction (SPE). Solid-phase extraction and LC-MS/MS analysis were performed in accordance with Weisser et al. [22]. Extraction was conducted using Bond Elut C18, 500 mg cartridges with a 10 mL reservoir (Agilent Technologies) placed on a vacuum manifold (IST Vacmaster, Biotage, Uppsala, Sweden). Prior to loading the samples, each SPE cartridge was preconditioned with 5.0 mL of methanol and 5.0 mL of MilliQ-water with a flow of approximately 1 mL/min. Following preconditioning, the diluted samples were transferred to the SPE cartridges at a flow of approximately 1 mL/min. The cartridges were then washed with 9.0 mL of H_2_O/MeOH (methanol) (75:25) solution with a flow of approximately 10 mL/min. The analytes were eluted from the cartridges by the addition of 5.0 mL of H_2_O/MeOH (20:80) solution at a flow of approximately 1 mL/min. The eluted extracts were collected in 10 mL glass tubes and evaporated to approximately 800 µL under a gentle stream of nitrogen at 50 °C. Following evaporation, 200 µL of 20% MeOH was added to the glass tubes, then vortexed and finally transferred to a clear Chromacol vial.

A binary 1290 Agilent Infinity Series system in combination with a binary 1100 Agilent HPLC pump was used for online clean up and chromatographic separation of steroids. For each sample, 100 µL was injected onto a C18 enrichment column (3.9 × 20 mm, 10 µm). The samples were then rinsed with an isocratic flow of 1 mL/min generated by the 1100 pump for two minutes with Milli-Q water/MeOH/formic acid (80:20:0.1 (*v*/*v*/*v*)). The steroids were retained on the column, while salts and proteins were washed into waste. Subsequently, the flow direction on the enrichment column was changed using a TTC switching valve, and steroids were transferred onto the analytical column (75 × 2.1 mm, 2.6 µm) with a guard column in front. The steroids were separated using a gradient elution with a flow of 0.3 mL/min generated by the 1290 pump. The mobile phases A and B were Milli-Q water/MeOH/FA (formic acid) in a ratio 80:20:0.1 and 100% MeOH, respectively. Mobile phase B was stepwise increased from the initial 10% to 99.5% at the end of the gradient elution at 12.5 min. Hereafter, the column was washed with 99.5% mobile phase B before re-equilibrating the column until 16 min. An AB SCIEX 4500 QTRAP mass spectrometer equipped with an atmospheric pressure chemical ionisation (APCI) tube 5 source was used for the detection and identification of the steroids. Data collection and processing were conducted using the Analyst v. 1.6.2 software package (AB SCIEX). Here, we present cortisol data, while estrogen and progesterone data are provided in Appendix A.

### 2.5. Statistical Analyses

A nonparametric Kruskall–Wallis test was conducted along with a pairwise post hoc Mann–Whitney U-test to test if the median cortisol concentration (pooling the days of collection) were different among individuals. Furthermore, a nonparametric Spearman rank test and both linear and polynomial regression (second degree) analyses were conducted for every cow and pooling all the cows on the cortisol concentration versus the cumulative number of warnings and electric impulses. See the Appendix A for more information about the statistical analyses and results.

## 3. Results

### 3.1. Cumulative Auditory Warning and Shocks

The number of warnings and electric impulses increased until around 12 June, after which they levelled off, around the time where the fence became fully virtual (Figure 3). This means that the cows had learned to appropriately respond to the auditory cues, reducing the number of both auditory cues and electrical pulses emitted (Figure 3).

### 3.2. Cortisol Concentrations in Manure

In total, 46 manure samples were collected from the five cows over the 18-day period and analysed afterward (Figure 4 and Appendix A). The cortisol concentrations in the manure varied from around 11 to 42 ng/g w/w. Except for one extreme value of 41.9 ng/g *w*/*w*, the concentrations were stable and did not vary much over time or between individuals (Figure 4 and Appendix A). No significant differences in median cortisol concentration were found among individuals (Kruskall–Wallis; *p* > 0.05), and no pairwise differences were found (Mann–Whitney U-test; all *p* > 0.05).

Moreover, neither the regression analyses (linear and polynomial) nor the Spearman rank test showed any significant trend for any of the individuals (all *p* > 0.05) (Figure 5 and Appendix A). All individuals had declining cortisol concentrations, except for cow #35-5 due to the extreme value. However, when removing the outlier, it resulted in a nonsignificant negative decline, similar to what was observed for the other four cows. When pooling the five individuals, the results remained unchanged, and no significant trend was found (*p* > 0.05) in the regression analyses (linear and polynomial) or the Spearman rank test (Appendix A).

## 4. Discussion

Cortisol is the classical mammalian stress hormone, produced and released to the blood stream by the adrenal cortex. It is regulated with negative feedback by the adrenocorticotropic hormone (ACTH) released by the pituitary gland in response to corticotropin-releasing hormone (CRH) [23]. Stress, acting through the nerve system, causes an elevation in ACTH, which, in turn, stimulates cortisol production and release from the adrenals. The stress response, promoted by cortisol, has widespread action, including the mobility of glucose and deamination of amino acids for gluconeogenesis. Cortisol also stimulates the mobilisation of fatty acids. All these actions serve the purpose of increasing the availability of quick energy to muscle and nerve tissue [24]. Following action, cortisol is then removed from the circulation by uptake and conjugation in the liver, followed by excretion via bile.

Cortisol is excreted via the bile as both glucuronide or sulphate metabolites, and the levels of free cortisol in bovine manure are therefore low. Consequently, a non-invasive method for assessing stress levels in cows is to analyse faecal cortisol metabolites (FCMs) as stress indicators [25]. Due to a 9–15 h delay in the gastrointestinal passage rate in cattle, FCMs are less prone to short-term variations during the day, and FCMs are therefore good indicators of adrenocortical activity and cortisol secretion over a longer period [26].

There are no standardised methods for analysing FCM, but two approaches are commonly used: analysing most or all cortisol metabolites using radioimmunoassays (RIAs) or chromatography, or deconjugating the FCMs and then analyse the total cortisol levels. In the present study, we used the latter approach, deconjugating all cortisol glucoronides and sulphates, thereby analysing the total cortisol level as: Cort_total_ = Cort_free_ + Cort_gluc_ + Cort_sul_. Using this approach, the total cortisol content ranged from 12 to 42 ng/g (w/w), which appears to be in the range previously reported for cattle. For example, in lactating cows, Ebinghaus et al. [25] reported an FCM range of 9–48 ng/g on organic dairy farms, and Rouha-Mülleder et al. [27] observed FCM concentrations at the farm level ranging from 9 to 48 ng/g. In beef bulls, Hernández-Cruz et al. [28] showed slightly lower FCM concentrations of 8–18 ng/g during different stages of the fattening period. Conversely, Bertulat et al. [29] showed FCM concentrations in lactating dairy cows ranging from 30 to 185 ng/g. Overall, the present results are in agreement with those of previous studies and indicate the use of the developed method for assessing total faecal cortisol levels.

If the virtual fence and the risk of electric shock directly applied to the nervous system of the cows were major stress factors, we would have found elevated levels of total cortisol levels in the manure of the cows with increasing number of warnings and shocks. Figure 5 clearly demonstrates that this was not the case, indicating that the virtual fence was not a major stress factor to the cows. Figure 5 also indicates some natural fluctuations in faecal cortisol, which were expected. In several mammalian species, it is well-established that cortisol secretin has a circadian rhythm, increasing during the night, to peak in the morning, while declining slowly during the day [30]. Based on the data presented in Figure 5, we speculated that the virtual fence was a minor or insignificant stress factor compared with the effects of the circadian rhythms and impacts from other natural fluctuations in the environment. In addition to this, natural biological variation may explain the high value of cow #35-5 [31,32].

### Considerations

Using cortisol analyses of manure in cattle may seem trivial. However, implementing this when using virtual fences in a year cycle, collecting fresh manure samples from the same individuals is one of the strengths of this study. Collecting manure requires surveys and human resources to ensure that the material is correctly collected from each individual while not stressing the animals, which was the reason for only testing 5 of the 12 cows in the present study. Cortisol is a marker of stress but cannot be used alone as a sure indicator of cows being stress-free. It is important to also include other assessments of welfare, such as behavioural analyses. The results of this study should be considered along with the findings of the behavioural analyses conducted by Aaser et al. [12], showing that the cows did not display any significant behavioural changes upon receiving an electric impulse. Hence, our results are supported by the behavioural studies conducted on the same animals.

## 5. Conclusions

There were no significant differences in the median cortisol concentration among individuals, and no significant trends were found over the period for any of the five cows in terms of exposure to auditory and electric cues. Furthermore, the cortisol concentrations did not significantly change over the study period for any of the five individuals. We therefore concluded that there is no indication that the use of virtual fencing impairs the welfare of the cows. In future studies, it would be interesting to extend the period of investigation and have a pretreatment and/or control group in order to be able to compare the cortisol levels found in this study. However, logistic constrains can make such experimental design difficult, as it is difficult to ensure constant environmental conditions (weather changes and necessity to move the cattle to different foraging areas). Thus, virtual fencing is a promising alternative to traditional physical electric fencing. Furthermore, we recommend the use of manure as a noninvasive measure of large grazers’ stress using deconjugation and LC-MS/MS analyses, to survey and understand animal welfare during the use of virtual fencing.

## Figures and Tables

**Figure 1 animals-12-03017-f001:**
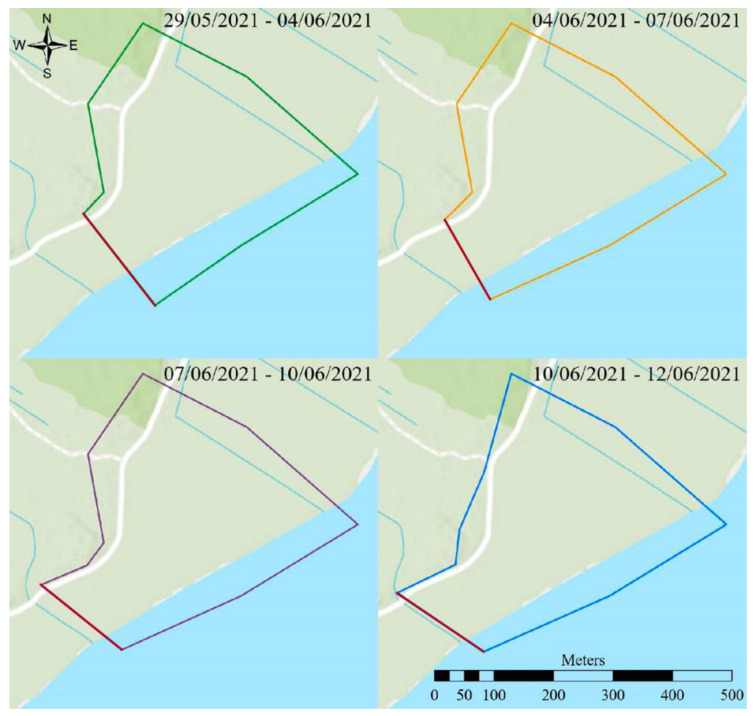
The semi-virtual enclosure on the island Fanø in the Danish Wadden Sea during 29 May to 12 June 2021. Only the southwestern fence line was completely virtual (red line); for the remaining borders, there was still a physical fence within the virtual border. The coordinates of the lower right corner of each map frame are 8°27′59″ E, 55°24′4″ N.

**Figure 2 animals-12-03017-f002:**
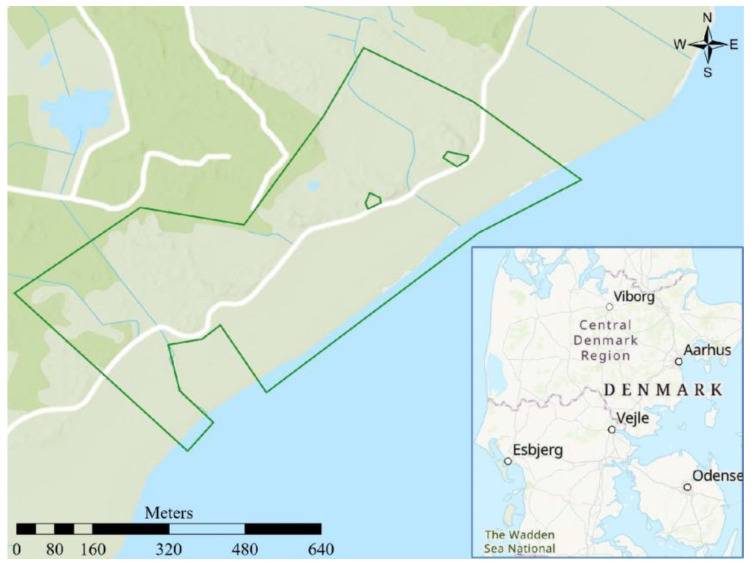
The virtual enclosure the days after the learning period (12 to 15 June 2021). Here, all borders were completely virtual. The coordinates of the lower right corner of the map frame are 8°28′9″ E, 55°23′43″ N.

**Figure 3 animals-12-03017-f003:**
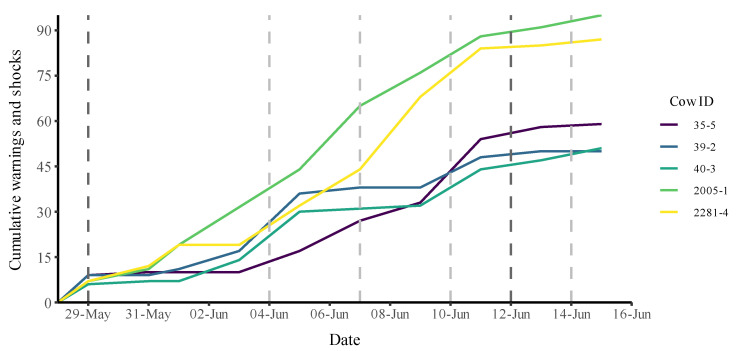
Cumulative auditory warnings and electric impulses during the study period of 29 May to 15 June 2021 where manure was collected from five animals for cortisol analyses. The virtual training started on 29 May; from 12 June onward, the fence was solely virtual. The vertical plot lines in represent significant changes in the virtual boundaries, with the darker lines indicating the beginning (29 May) and end (12 June) of the virtual training period.

**Figure 4 animals-12-03017-f004:**
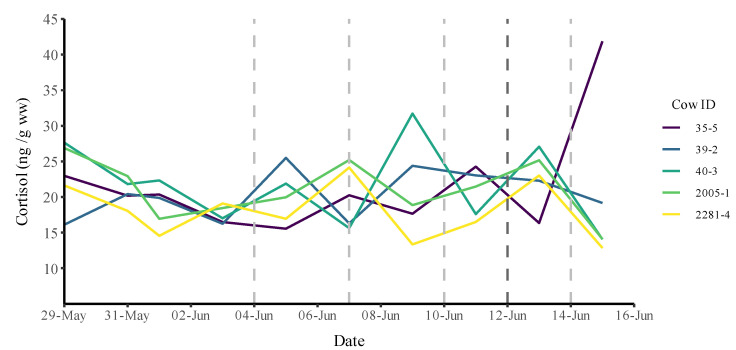
Cortisol concentrations in manure from the five cows sampled in the period of 29 May to 15 June. Training for virtual fencing started on 29 May; from 12 June onward, the fence was solely virtual. Coloured lines indicates each cow. Vertical lines: changes in fence position.

**Figure 5 animals-12-03017-f005:**
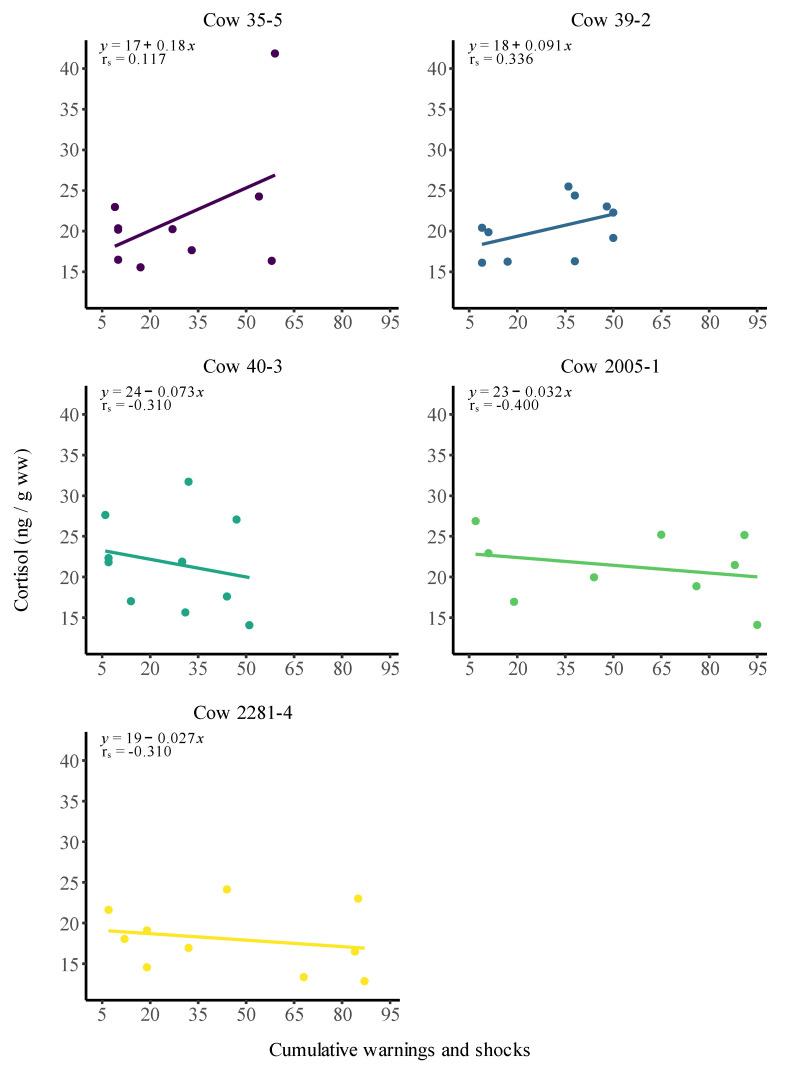
Linear regression of cortisol concentration in relation to the cumulative number of auditory warnings and electric impulses received for each individual during the study period of May 29 to June 15. The regression equation and Spearman correlation coefficient (r_s_) are given in each plot. New colour for each cow.

## Data Availability

All data are found in the Appendix A.

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
