# Peer review of "Cortisol in Manure from Cattle Enclosed with Nofence Virtual Fencing"

_animals, 2022, doi:10.3390/ani12213017_

Round 1

Reviewer 1 Report

The authors presented a manuscript regarding the effects on cow welfare of the use of virtual fencing. This is an important topic because the application of this system could improve the cow management. In detail, the authors evaluated the welfare of the cows by the assessment of the cortisol concentrations in the manure. 

The topic is interesting, but I would like to provide some comments because the manuscript is not clear enough. In particular, materials and methods should be implemented in order to clarify the method applied to select the sample, the period of investigation and the area under study. I have no specific comments about the results and the discussion because it is important to have more information about the materials and methods. Therefore, I suggest the authors a major revision. 

1.      The Introduction should be improved and rewritten in some parts. Different works are mentioned in the introduction but there is no discussion on the state of the art that leads to identify the research gaps. Moreover, many sentences are not supported by the literature. I suggest: 

  • Adding references in lines 52-53, 70, 73, 77-79, 88-90. 
  • Explaining what are the main public concerns found in the literature; 
  • Discussing the main results obtained in recent research studies; 
  • Identifying the gaps found in the literature  
  • Adding the novelty of the study and, therefore, reformulating the aims of the work.  

2.      The authors declared in the aim of the work was to assess the effect of a virtual fencing system on the welfare of twelve Angus cows, but in the materials and methods they reported that due to logistics only five of the twelve cows were used in this particular experiment, as collecting samples from all twelve individuals would have increased the risk of mixing up the samples. All twelve cows were pregnant during the experiment. It is not clear why the author reported that the study was carried out for 12 cows, but actually the sample was composed only by 5 cows. However, it is not clear how the authors selected these five cows in the group of twelve cows. The sample is composed of few elements. I could accept a small sample but it is necessary to motivate with scientific evidence how the sample was selected. Moreover, I suggest removing from the text the sentence “as collecting samples from all twelve individuals would have increased the risk of mixing up the samples” because it is too vague.  

3.      The total area of the project was 121 ha, the maximum area used during this particular experiment was limited to 35 ha. Please clarify, the method applied to select this small area in the research study (for example specific features of the area under study). Moreover, it is not clear the reason for changing the virtual border of the fence line.  

4.      The cow behaviour and welfare could be influenced by many factors (for example climatic conditions). Please, clarify the method applied to select the days during the period of investigations. 

5.      Please, reversing the order of the section Materials and Methods: 2.1 Location and study design; 2.2 The virtual fencing system Nofence©; 2.3 Sampling  

Author Response

REV 1:

  • The authors presented a manuscript regarding the effects on cow welfare of the use of virtual fencing. This is an important topic because the application of this system could improve the cow management. In detail, the authors evaluated the welfare of the cows by the assessment of the cortisol concentrations in the manure. Thank you very much for this positive recap.
  • The topic is interesting, but I would like to provide some comments because the manuscript is not clear enough. We are very happy for that! In particular, materials and methods should be implemented in order to clarify the method applied to select the sample, the period of investigation and the area under study. Thank you very much. We will consider that along that the editor would like us to reduce overlap with previous/other publications so it may not be entire possible. I have no specific comments about the results and the discussion because it is important to have more information about the materials and methods. Therefore, I suggest the authors a major revision. We appreciate this!
  • The Introduction should be improved and rewritten in some parts. Different works are mentioned in the introduction but there is no discussion on the state of the art that leads to identify the research gaps. Moreover, many sentences are not supported by the literature. I suggest:
  • Adding references in lines 52-53, 70, 73, 77-79, 88-90. Thank you very much! This is now corrected.
  • Explaining what are the main public concerns found in the literature; Thank you very much! This is now corrected.
  • Discussing the main results obtained in recent research studies; Thank you very much! This is now corrected.
  • Identifying the gaps found in the literature Thank you very much! This is now corrected.
  • Adding the novelty of the study and, therefore, reformulating the aims of the work. Thank you very much! This is now corrected.

  1. The authors declared in the aim of the work was to assess the effect of a virtual fencing system on the welfare of twelve Angus cows, but in the materials and methods they reported that due to logistics only five of the twelve cows were used in this particular experiment, as collecting samples from all twelve individuals would have increased the risk of mixing up the samples. All twelve cows were pregnant during the experiment. It is not clear why the author reported that the study was carried out for 12 cows, but actually the sample was composed only by 5 cows. Due to logistics concerns making it impossible to ensure that the manure could be traced back to each individual as it requires 24H surveillance by 5-10 individual observers due to the size of the area. The 5 cows were simply randomly collected blindly from the brutto list of 12 cows. This is now clarified in the revised ms. However, it is not clear how the authors selected these five cows in the group of twelve cows. The sample is composed of few elements. I could accept a small sample but it is necessary to motivate with scientific evidence how the sample was selected. These were all randomly selected by the farmer and to ensure man-power so that each manure sample could be allocated back to the individual cows with 100% certainty. This is now added to the ms. Moreover, I suggest removing from the text the sentence “as collecting samples from all twelve individuals would have increased the risk of mixing up the samples” because it is too vague. This is now done. Thank you very much!
  2. The total area of the project was 121 ha, the maximum area used during this particular experiment was limited to 35 ha. Please clarify, the method applied to select this small area in the research study (for example specific features of the area under study). This is again due to logistics. We camera-monitored this area the first 2 weeks. For a larger area it would not be possible to monitor behavior, collect manure a.o. The area was chosen from a logistic point with access to power, roads and research facilities (housing, labs a.o.). This is added to the revised ms. Moreover, it is not clear the reason for changing the virtual border of the fence line. Changing the fence line of the virtual border was conducted step-wise to ensure a smooth learning curve. This is now added to the revised ms.
  3. The cow behaviour and welfare could be influenced by many factors (for example climatic conditions). Please, clarify the method applied to select the days during the period of investigations. The 14-days were selected to ensure a short window of stable weather with respect to wind, participation and food access. This is now added to the revised ms.
  4. Please, reversing the order of the section Materials and Methods: 2.1 Location and study design; 2.2 The virtual fencing system Nofence©; 2.3 Sampling. This is done in the revised ms. Thank you very much.

Author Response

REV 2:

Review for Manuscript ID: animals-1969744

  • Summary: This manuscript adds to the growing knowledge about utilizing virtual fencing for grazing animals. Specifically, this paper evaluates some of the physiological aspects of the potential stress involved with the practice. Some obvious limitations are involved with the project and with the technology, but this is a good proof of concept scale project. Thank you very much for this recap.
  • General comments about the article: Was the training area also utilized as the fully virtually fenced area post-training? This was not clear to me in the materials and methods, so I believe it may need to be adjusted. We are very sorry that this is not clearer. The training area is encompassed in the larger utilized area during the 2 year+ project. This is clarified in the revised manuscript. Only 2 of the 10 samples per cow were taken during the totally virtual fence phase of the trial period, do you believe that confounded or affected the variability of cortisol? We believe that all parameters that could affect the cortisol was accounted for including access to food while temperature and weather was stable. We did our best to have as many manure samples from as many individuals as equally distributed as possible but that was really a challenge (please see above reply to REV 1). Looking at the data including variability and the lack of any kind of development (increase/decrease) in the dataset we believe that the design is reasonable. Form a practical point of view, the cows also had to leave the training area to utilize the larger 65 ha ecosystem. Do you believe any of the increases in warnings and shocks during the training period were due to lack of forage or other nutritional aspects? To us it looks that none of the cows were affected by the virtual fence compared to classical physical fences. Except during a single occasion where they escaped within the first 30 min and all got a single shock. Based upon figure 5, are all of the cortisol concentrations, other than Cow 35-5 with her outlier value, declining as discussed in the Results? Overall a very interesting paper, but the logistical challenges which necessitated such a short virtual only sampling period seems to present some questions with the confidence upon which the conclusions are drawn and presented. We agree that this is not a large-scale and very thorough study therefore having limitations. It would have required many more resources and approval of a larger sample size for the experiment. Thank you for the perspectives and questions raised.
  • General comments about the review: The use of virtual fencing is a fairly new production concept which has some tremendous applications around the world. Since it is an emerging field, there are somewhat limited resources to utilize for the review, which I believe was well done for this manuscript. Thank you very much. Again, we are thankful for the perspectives raised by REV 2 and we are aware that this is not a large-scale and ideal study due to many reasons, among which the number of animals used, number of samples and time-scale are among these as pointed out by the reviewer.
  • Specific comments: On Line 23, add a comma after “cues”, remove “that” and add an s to “encourage”. On Line 26, add “cattle” or “cows” after mature. Line 28, change “of” to “for”. Thank you very much. This is now done.
  • On Line 40, remove “of”. On Line 43, change “of” to “for”. Line 53, add been after “has”. On Line 65, add receiver or unit following “GPS”. For the sentence beginning on Line 68 and ending on Line 70, revise as it does not make sense as written. On Line 79, change “determine” to “determining”. In the Materials and Methods, regarding the Cortisol Analysis, check consistency of “mL” vs. “ml”. Thank you very much. This is now corrected.

Also in the Cortisol Analysis, be sure to define “MeOH” and “FA” prior to using them as abbreviations. In Line 171, change “A” to “The”. On Line 179, I believe “ml or mL” needs to be added between “5.0” and “methanol”. On Line 242, change “Cow 35-1” to “Cow 35-5”. Thank you very much. This is now corrected.

Reviewer 3 Report

Review “Cortisol in manure from cattle enclosed with Nofence virtual fencing”

The authors Sonne et al. describe a study in 12 cattle that were enclosed in an area that had traditional electric fencing, which was gradually replaced by virtual fencing whereby cattle received an auditory stimulus if they approached the virtual border followed by an electric stimulus if they were at risk of crossing the border. Faecal sample were collected over the testing period to determine cortisol metabolites, which is linked to stress levels. The authors did not find elevated cortisol levels over time and concluded that a virtual fence does not impair the welfare of the tested cattle.

The authors already discuss the limitations of the study, such as extension of the period of investigation and inclusion of a pre-treatment or control group, which are valid points. Numbers are small and individual variations make interpretation difficult.

My main concern are the data in the supplementary file, which are either erroneous or misinterpret them. All 5 cattle were in an area with a traditional fence and all received three shocks, which is an odd coincidence but may be true. According to the text, virtual fencing started on 29 May so why is it stated in the table that it was traditional fencing. Should the observation and sampling not have started on 28 May when there was only a physical fence? I cannot follow the cumulative shocks and auditory stimuli as they do not match up. For example, cow 35-5 received 3 shocks on 29 May but no further shocks; the cumulative shocks however were 8 at the end of the study period. Does it mean it received shocks on the days that are not covered in the table? They had six warnings on 29 May and five on 5 June by which time the cumulative stimuli were already 13 but should have been 11. Again, did they receive warnings in the days that were not in the table? Without using any statistics one would simply look at number of warnings/ shocks and cortisol levels the day after (assuming that the effect is measured 24 hours later) but I am not sure whether they figures are correct.

The authors explain the choice of their statistical test in the supplementary file. Another analytical approach, run in parallel, would have been to keep cattle in a physical enclosure only for e.g. 2 days and analyse a sample on day 2 as a baseline sample. A second sample would then be analysed when the fencing is entirely virtual, and pair-wise comparison between physical and virtual fence would be possible. Given the fluctuations in values during sampling periods it is probably advisable to measure faecal samples several times during the day rather than a single sample, which is logistically even more complicated but would establish a mean or median value, which could then be compared as a summary statistic value. Was this not considered? Why was sampling not started much earlier when the cattle were physically enclosed to see the variation at that time.

The discussion does not mention the single high value in cow 35-5. Is there any explanation? Could this be pregnancy-related?

Faecal cortisol may be a measure of stress in cows but there are also studies that demonstrate that not all disturbances cause an increase in faecal cortisol (e.g. Möstl et al. 2002). Maybe it would be useful to go into more detail why the study should be interpreted in the context of the behavioural study.

Fig 1. It would be useful to draw the “virtual” border in a different colour to make it clear what the south western border is.

Author Response

REV 3:

  • Review “Cortisol in manure from cattle enclosed with Nofence virtual fencing”.
  • The authors Sonne et al. describe a study in 12 cattle that were enclosed in an area that had traditional electric fencing, which was gradually replaced by virtual fencing whereby cattle received an auditory stimulus if they approached the virtual border followed by an electric stimulus if they were at risk of crossing the border. Faecal sample were collected over the testing period to determine cortisol metabolites, which is linked to stress levels. The authors did not find elevated cortisol levels over time and concluded that a virtual fence does not impair the welfare of the tested cattle. Thank you very much for this nice wrap-up!
  • The authors already discuss the limitations of the study, such as extension of the period of investigation and inclusion of a pre-treatment or control group, which are valid points. Numbers are small and individual variations make interpretation difficult. Thank you for seeing these points.
  • My main concern are the data in the supplementary file, which are either erroneous or misinterpret them. Thank you very much for these questions and points raised.
    • All 5 cattle were in an area with a traditional fence and all received three shocks, which is an odd coincidence but may be true. All cows escaped on the first day and received the maximum of three shocks before being registered as escaped and no shocks were received after the cows were returned to the enclosure for the rest of the day.
    • According to the text, virtual fencing started on 29 May so why is it stated in the table that it was traditional fencing. Should the observation and sampling not have started on 28 May when there was only a physical fence? The virtual fencing was first started in the evening of May 29th, thus, the fencing at the time of the manure samples was still only traditional fencing. This has now been clarified in both the caption and by adding an additional row for each cow on May 29th.
    • I cannot follow the cumulative shocks and auditory stimuli as they do not match up. For example, cow 35-5 received 3 shocks on 29 May but no further shocks; the cumulative shocks however were 8 at the end of the study period. Does it mean it received shocks on the days that are not covered in the table? They had six warnings on 29 May and five on 5 June by which time the cumulative stimuli were already 13 but should have been 11. Again, did they receive warnings in the days that were not in the table? Without using any statistics one would simply look at number of warnings/ shocks and cortisol levels the day after (assuming that the effect is measured 24 hours later) but I am not sure whether they figures are correct. Yes, the cumulative values also include shocks and warnings received on days between the collection of manure samples. This is now explained in the table caption.
  • The authors explain the choice of their statistical test in the supplementary file. Another analytical approach, run in parallel, would have been to keep cattle in a physical enclosure only for e.g. 2 days and analyse a sample on day 2 as a baseline sample. A second sample would then be analysed when the fencing is entirely virtual, and pair-wise comparison between physical and virtual fence would be possible. Given the fluctuations in values during sampling periods it is probably advisable to measure faecal samples several times during the day rather than a single sample, which is logistically even more complicated but would establish a mean or median value, which could then be compared as a summary statistic value. Was this not considered? Why was sampling not started much earlier when the cattle were physically enclosed to see the variation at that time. Thank you very much for these very valid points. We agree on all this and the answer may seem a bit blunt. The work associated with collecting these (rather few) samples was tremendous given our small team and the logistically constraints and time budget associated with the efforts to ensure that each manure samples was linked with each individual. Basically, we would have loved samples each 2nd hour during several month from each of the 12 cows, however, this was simply not possible given the trial was run by two persons for the first weeks. In addition, and more importantly, the amount of grass was relatively little in the physical enclosure, and therefore only allowed a limited time before replacing the physical fence with virtual ones.
  • The discussion does not mention the single high value in cow 35-5. Is there any explanation? Could this be pregnancy-related? Basically, we do not know this. Mostly likely, the value is due to biological variation. Regarding gestation, cortisol does not seem to vary throughout the period except for cows having twins making cortisol to increase 2 days pre-partum (https://pubmed.ncbi.nlm.nih.gov/8791850/).
  • Faecal cortisol may be a measure of stress in cows but there are also studies that demonstrate that not all disturbances cause an increase in faecal cortisol (e.g. Möstl et al. 2002). Maybe it would be useful to go into more detail why the study should be interpreted in the context of the behavioural study. This is absolutely true. The Möstl et al. 2002, however, uses a different analytical method which make it problematic to compare the studies. In addition, as pointed out by REV 3 this study does not contain a wealth of data including manipulative ones so we feel it’s not appropriate to dive into these perspectives in great detail. Despite of this, we have mentioned Möstl et al. 2002 in the Discussion.
  • Fig 1. It would be useful to draw the “virtual” border in a different colour to make it clear what the south western border is. This has now been done.

Reviewer 4 Report

very interesting work
well-processed results and relevant conclusions

Author Response

We thanks.

Round 2

Reviewer 1 Report

Dear Author, 

I accept the manuscript in the present form. 

Author Response

We thanks the reviewer!

Reviewer 3 Report

This manuscript has been improved considerably and I have no major concerns but a few recommendations:

- I am content with your explanation why the design was chosen and the lack of staff to sample more frequently. It may be advisable to add this to the discussion as it may also be questioned by readers once it is accepted for publication.

- I still think that too little emphasis is placed on the outlier. The authors added a sentence on line 319-321 but as it is stated it does not explain the INCREASE in faecal cortisol. I would reword the sentence to say that biological variation may also increase faecal cortisol, which may be the most likely explanation for the outlier.

- The added sentence in line 96-97 makes little sense to me as it stands. Should it be: "Here, we hypothesise that using virtual fencing does not compromise Angus cow's wellbeing measured as manure cortisol concentrations."?

Author Response

REV 3:

This manuscript has been improved considerably and I have no major concerns but a few recommendations:

- I am content with your explanation why the design was chosen and the lack of staff to sample more frequently. It may be advisable to add this to the discussion as it may also be questioned by readers once it is accepted for publication. Thank you very much for this valid point! This is now added to Considerations in the Discussion.

- I still think that too little emphasis is placed on the outlier. The authors added a sentence on line 319-321 but as it is stated it does not explain the INCREASE in faecal cortisol. I would reword the sentence to say that biological variation may also increase faecal cortisol, which may be the most likely explanation for the outlier. Thank you very much for this valid point! We of course agree and have done this.

- The added sentence in line 96-97 makes little sense to me as it stands. Should it be: "Here, we hypothesise that using virtual fencing does not compromise Angus cow's wellbeing measured as manure cortisol concentrations."? Thank you very much! We have revised this sentence.